# Long-term changes in the Juvenile Sockeye Salmon Rearing Capacity of the Chignik Lakes Watershed

Cirque Gammelin [ID]*, Daniel E. Schindler*

School of Aquatic and Fishery Sciences, University of Washington, Seattle, Washington, United States of America

* cirqug@uw.edu (CG); deschind@uw.edu (DES)

## Abstract

Freshwater ecosystems respond rapidly to perturbations in climate, geomorphology, and population abundances. For migratory species in interconnected habitat networks, local habitat conditions can control the productivity of individual populations. Asynchronous variation in habitat quality can simultaneously stabilize ecological processes at broad scales but also complicate understanding of ecosystem dynamics. We investigated habitat-specific trends in indicators of the rearing capacity for juvenile sockeye salmon in a remote watershed in Alaska over the last ~60 years. The motivation of this effort was to understand if the collapse of the local salmon fishery in 2018 could be traced to changes in habitat quality within the nursery watershed. Our analyses describe high variability in the habitat conditions across both spatial and temporal scales, yet do not suggest a decline in the overall sockeye salmon rearing capacity of the watershed. We observed increasing maximum water temperatures in a shallow lake but more stable conditions in a deep lake, an improvement in zooplankton prey resources in the deep lake, and increased juvenile sockeye salmon growth rates throughout the watershed. Although we detected no long-term decline in rearing habitat quality, there was a decrease in juvenile sockeye salmon abundance from 2013–2016, suggesting high early life stage mortality prior to the period of juvenile rearing leading up to the fishery collapse.

## Introduction

As global climate change continues, understanding ecological responses to changing environmental conditions in lakes has utility for developing successful adaptive management strategies for key resources and the ecosystem services they provide. Lakes are inherently dynamic ecosystems, whose physical structure is controlled by interactions between climatic drivers and geomorphological characteristics of their watersheds [1]. The system of chemical, physical, and biological processes which filter climate variables in aquatic ecosystems makes lakes respond in complex and

**Data availability statement:** All data files will be available upon publication from the Dryad database (DOI: 10.5061/dryad.8sf7m0d29). Reviewers may access the database using the following link: http://datadryad.org/share/ LINK_NOT_FOR_PUBLICATION/K-tkL1QGRfmu BSgfE99eNyCplOsPO4dwMw8wdRnv1pY DOI: https://doi.org/10.5061/dryad.8sf7m0d29.

**Funding:** Funding was provided by the Gordon and Betty Moore Foundation (CG,DS), the Chignik Regional Aquaculture Association (DS), the Fishery Disaster Relief Program of the Pacific States Marine Fisheries Commission (DS), and the School of Aquatic and Fishery Sciences at the University of Washington (CG, DS). The funders had no role in study design, data collection and analysis, decision to publish, or preparation of the manuscript.

**Competing interests:** The authors have declared that no competing interests exist.

often unpredictable ways to climate variation [2,3]. The responses of lake biota to changing physical and chemical drivers can feed back to alter direct responses to changing climate through phenomenon such as recruitment variation of important species [4] which can produce changes in the strength of trophic cascades [5].

Fisheries management is complicated by uncertainties in both density-dependent feedbacks within exploited populations, and in the dynamics of the habitat used by organisms to complete their life cycles. For migratory species, understanding changing ecological constraints on productivity is particularly difficult, as they use a variety of habitats over the course of their lives [6]. Because regional and local habitat conditions can respond asynchronously to climate forcing, habitat connectivity enables individuals to migrate among habitats to exploit changing growth and survival opportunities and avoid stressful conditions as habitats respond to regional climate forcing [7]. Thus, habitat complexity encountered by exploited species can simultaneously stabilize population dynamics and complicate understanding ecosystem responses to changing climate, thereby challenging the development of prescriptive management strategies [8–10].

Pacific salmon (*Oncorhynchus* spp.) fisheries are supported by heterogeneous freshwater habitats, providing economic and cultural ecosystem services through subsistence and commercial harvest [11]. Maintained by strong natal homing [12–14], salmon populations exhibit a wide range of life histories among genetically distinct populations within watersheds [15,16]. Sockeye salmon (*Oncorhynchus nerka*) are particularly reliant on lakes and their watersheds for spawning and juvenile rearing habitat. Marine survival of sockeye salmon is positively correlated to the size and body condition of out-migrating smolts [17–20]. If rearing habitat conditions are degraded, juvenile sockeye salmon can experience poor growth performance, decreasing chances of ocean survival and ultimately population recruitment. Therefore, dynamics of exploited sockeye salmon populations are ultimately responsive to both freshwater and marine habitat conditions, as well as density-dependent responses to harvest [21].

Situated on the Alaska Peninsula, the Chignik Lakes watershed provides spawning and rearing habitat for multiple genetically distinct populations of sockeye salmon [22] which have supported a vibrant commercial fishery for over a century, and are a crucial subsistence resource for local communities [23]. Chignik sockeye salmon stocks are aggregated into early run (May-July) and late run (July-September) populations for management purposes. Commercial and subsistence harvest of both runs is managed by the Alaska Department of Fish & Game (ADF&G), who operate a weir on the lower Chignik River to enumerate and provide stock assignment of returning adult fish. Their management strategy seeks to maximize fishing opportunity while achieving escapement goals set to maximize long-term yield from the stocks [24].

In 2018, Chignik sockeye salmon experienced a precipitous decline in the number of returning adults (Fig 1), resulting in several years of painful closures of the commercial fishery. Concern over the future sustainability of the Chignik sockeye salmon fishery motivated the ADF&G to propose implementing a new management strategy to account for a hypothesized regime shift within the freshwater rearing habitat of the

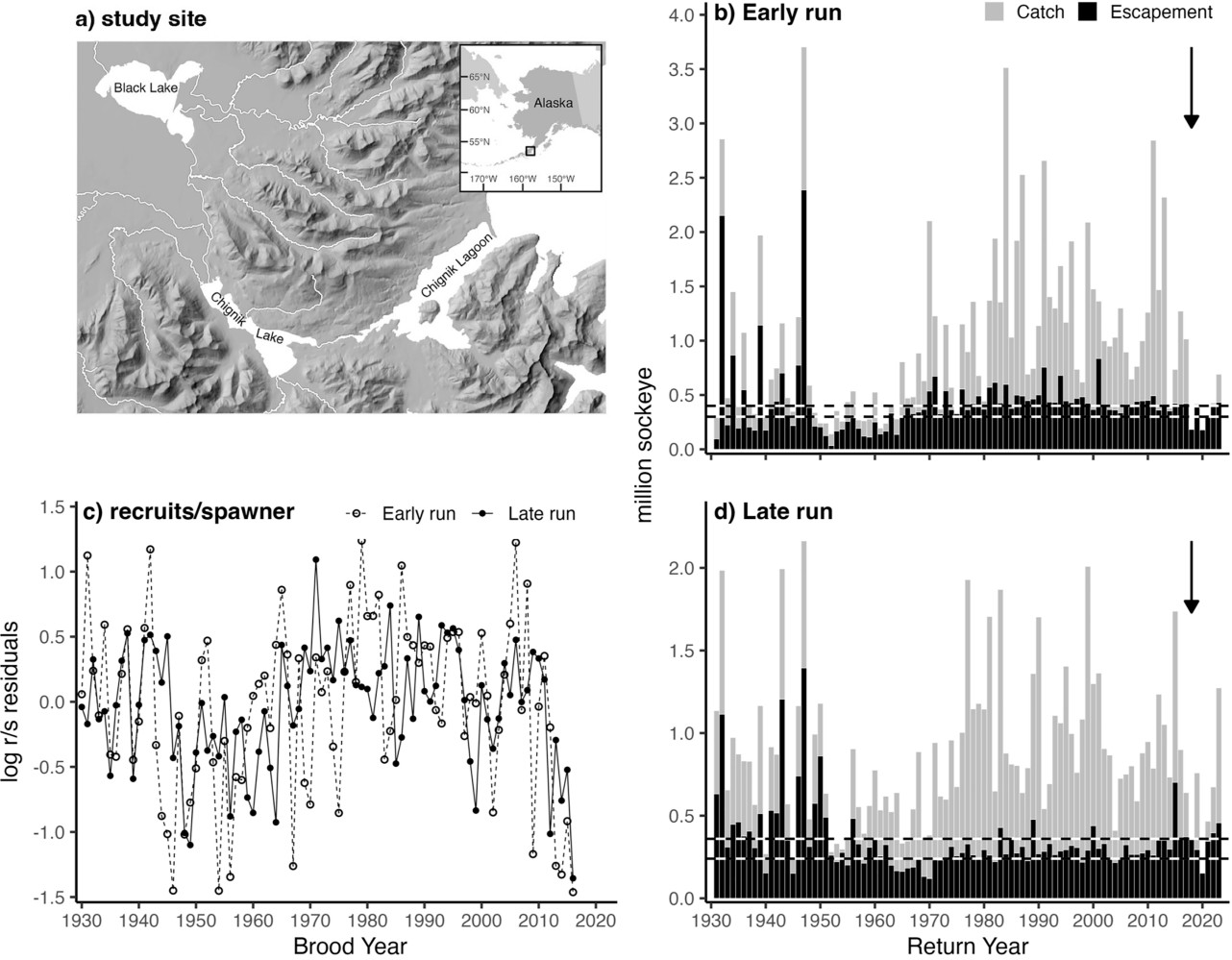

**Fig 1. Map of the Chignik lakes watershed (a).** Adult sockeye salmon returns to the Chignik lakes watershed by return year, for both early **(b)** and late runs **(d)**. Horizontal dashed lines show 2023 lower (300,000 early run, 240,000 late run) and upper (400,000 early run, 360,000 late run) sockeye salmon biological escapement goals for early **(b)** and late **(d)** runs set by ADF&G. Black arrows indicate the 2018 sockeye salmon stock collapse. Log recruit/spawner residuals by brood year, for both the early and late runs **(c)**. Basemap data sourced from USGS National Elevation Dataset, USGS National Hydrography Dataset, and Natural Earth. Sockey salmon return data provided by the Alaska Department of Fish & Game.

system [24]. Citing geomorphological changes [25,26] and climate as controls on habitat quality, ADF&G hypothesized that the watershed is no longer able to support historic abundances of juvenile sockeye salmon. While the proposed management strategy was not implemented due to a lack of empirical evidence of declining habitat quality, a clear consensus on the cause of the 2018 collapse has not yet been reached. Without a comprehensive analysis of temporal trends in the habitat quality of the Chignik watershed, it remains unclear how freshwater rearing capacity has changed, and whether potential changes justify altering management strategies to accommodate degrading freshwater habitat conditions.

With diverse and complex life histories, it can be challenging to identify and quantify the mechanisms behind sockeye salmon stock dynamics. Factors such as freshwater habitat quality, oceanic conditions, recruitment success, and harvest all contribute to the temporal stability of individual stocks [27,28]. While this study focuses on the hypothesized declines in freshwater rearing capacity in the Chignik watershed, it is possible multiple mechanisms contributed to the 2018 stock

collapse. For example, an emerging hypothesis asks if extreme early winter flood events may have scoured sockeye salmon redds, resulting in poor egg and fry survival in the years preceding the collapse.

We used several decadal-scale datasets to characterize long-term trends in lake thermal conditions, zooplankton community, planktivorous fish community, and juvenile sockeye salmon growth performance to quantify evidence for changes in the rearing capacity of the Chignik lakes watershed. Comparing trends in rearing capacity between interconnected shallow (Black Lake) and deep (Chignik Lake) nursery lakes, we investigated asynchronous changes to habitat and population dynamics in a heterogeneous watershed. In particular, we focused on ecosystem shifts during the juvenile sockeye salmon rearing years (2013–2016) preceding the (2018–2021) stock collapse, corresponding to the typical 1–2 years of freshwater residency and 2–3 years of marine residency of sockeye salmon [23,29]. We organized our analyses of historical data collected over the last several decades to assess the evidence that the growth and rearing conditions experienced by juvenile sockeye salmon in this ecosystem have declined in recent years. In particular, we asked the following questions:

1) How have summer thermal conditions changed within the two dominant nursery lakes of the Chignik watershed?

2) Has the zooplankton community structure of Chignik and Black lakes changed, and do these changes reflect responses to top-down predation effects or bottom-up environmental effects?

3) How have abundances of resident planktivorous fishes that compete with juvenile sockeye salmon changed in ways that would affect the growth and survival of juvenile sockeye salmon?

4) What were the lake-specific growth responses of juvenile sockeye salmon to habitat change?

## Methods

### Study area

The Chignik Lakes watershed (N56°16' W158°50') on the Alaska Peninsula drains southward through two lakes and a large semi-enclosed estuary into the Gulf of Alaska. Situated in the upper watershed, Black Lake is large (35 km$^2$) and shallow ($\overline{X}$= 1.5 m), sitting in a low tundra depression on the northern slope of the Alaska Peninsula. Black Lake is relatively warm and turbid, with high primary and secondary production during the summer growing season. Chignik Lake, located downstream in a glacial valley, is smaller (22 km$^2$) and deeper ($\overline{X}$= 64 m). Persistent winds keep both lakes well mixed during the ice-free seasons and thermal stratification is rare in both lakes. Black Lake is fed by a primary tributary, the Alec River, and drains into Chignik Lake via the Black River. Chignik Lake empties into the Chignik Lagoon through the short Chignik River (Fig 1a).

Anadromous sockeye salmon return to spawn in the watershed in two distinct periods. From May to mid-July, 'early run' adults enter the watershed, migrating to tributaries of Black Lake to spawn. Early run juveniles rear primarily in Black Lake, growing rapidly before emigrating to sea as smolts after one year of growth. From mid-July to September, 'late run' adults spawn in tributaries and beaches of Chignik Lake and the Chignik River. Offspring of late run populations grow slower in Chignik Lake, typically rearing for two years before leaving the watershed as smolts. The collapse of the Chignik sockeye salmon fisheries in 2018 and subsequent years was particularly acute for the early run stocks that spawn in Black Lake, though the number of recruits-per-spawner for both stocks have declined notably since the 2014 brood year (Fig 1c).

Natural geomorphological and hydrological shifts during the 1970s resulted in notable changes to the habitat structure of the watershed. Downstream movement of the confluence of the dynamic West Fork River and the Black River, eliminated a key source of sediments that had maintained the elevation of the outlet of Black Lake [30,31]. Subsequent erosion resulted in a~40% decline in the volume of Black Lake, reducing the per capita volume of rearing habitat available to juvenile sockeye salmon. Black Lake volume has remained stable since the 1980s [32].

## Lake temperature

Water temperatures in Chignik and Black Lakes were compiled from multiple sources and summarized into monthly summer averages spanning 1929–2023. Data from 1929–1932 were obtained from historic logbooks from the Federal Fish and Wildlife Commission Chignik Field Station, accessible via the University of Washington (UW), School of Aquatic and Fishery Sciences archives. Methods and site locations from this period were unavailable. Surface water temperatures between 1955–1967 and 1990–2023 were recorded by the UW Alaska Salmon Program (ASP) from routine limnological surveys. Sampling occurred approximately every 10 days from June-September at two sites in Chignik Lake and three sites in Black Lake. Between 1995–2023, Yellow Springs Incorporated and CastAway sondes were deployed during limnological sampling at two sites in Chignik Lake to create vertical temperature profiles of the water column. As Black Lake is shallow and well mixed, depth profiles were not collected, and we simply summarized surface temperature measurements. From 2013–2023, Onset Computers pressure transducers were deployed year-round at two sites in Black Lake, recording hourly water temperature and lake levels. Loggers were anchored to the lake bottom and recovered and redeployed annually. Lake level data were corrected for changes in atmospheric pressure monitored by parallel pressure transducers at the lake edge.

Temperature measurements were compiled into a single dataset, and mean monthly temperatures were calculated for each year data was collected. In Black Lake, all temperature readings were used in our analysis. In Chignik Lake, stratification is rare due to sustained high winds. To reduce effects of occasional weak stratification events, only measurements take at 10 m depth or less were used in analysis.

Maximum and mean monthly water temperatures were calculated for both Black and Chignik Lakes in all available years. Only June, July, and August temperatures were used in analysis to represent the summer growing season. Data were separated into three periods, 1925–1950, 1951–1989, 1990–2023, and plotted by lake. Due to variation in sampling methodology and relatively few samples during the first two periods, a linear regression of trends in monthly maximum and mean temperatures was applied only to the most recent period. Significance of the slope was assessed using a two-tailed t-test with $\alpha = 0.05$. Regression results are reported as slope $\pm$ SE, t-statistic with residual degrees of freedom, $R^2$, and two-sided p-value (S1 Table, S2 Table). All statistical analyses were performed in R [33].

## Atmospheric temperature

Mean monthly atmospheric temperatures were used to reflect local climate variation in the watershed. Mean monthly air temperatures were obtained from the Scenarios Network for Alaska Planning (SNAP, <www.snap.uaf.edu/>). These data are derived from Climate Research Unit (CRU) dataset CRU TS 4.08, downscaled to 2 km grid cells. Temperature data are available from 1901–2023. A single grid cell (56°45′88″N, 158°99′77″W) centered on Black Lake was used to represent thermal conditions of the watershed following Griffiths et al. 2014 methodology [1].

## Zooplankton

Zooplankton samples were collected from Chignik Lake and Black Lake approximately every 10 days from June-September from 1961–2023. Plankton were sampled with 40 m vertical tows from two sites in Chignik Lake and 20 m horizontal surface tows from three sites in Black Lake. The net was retrieved at approximately 0.5 m/s by hand. To ensure vertical tows in windy conditions in Chignik Lake, a 6.8 kg weight was attached to the bottom of the net. In Black Lake, a float was attached to the mouth of the net to maintain surface contact. Over the course of the dataset, net mesh size and mouth opening size varied. Beginning in 2006, a conical net with 0.5 m wide mouth opening and 247 μm mesh size was used. Because net efficiencies of historical sampling gear were not estimated, we focus our analyses entirely on plankton community composition in this paper when comparing recent data to those collected before 2006. The same net was used after 2006 enabling the quantification of changes in estimated zooplankton densities from 2006–2023. Zooplankton

were preserved in 50–70% ethanol. Consecutive sub-samples were obtained with a Gilson pipette and enumerated under a dissecting microscope until approximately 500 individual zooplankters were counted. Zooplankton were identified as taxa-specific densities, and we do not have species-specific counts or data on the life stage of the organisms.

Because of unrecorded changes in net size and incomplete density units in historical datasets, we were unable to directly compare zooplankton densities among all periods. Analysis of zooplankton community composition was conducted in two steps on the four dominant taxa present in the watershed (*Daphnia, Bosmina,* calanoid copepods, cyclopoid copepods). A proportional composition plot was used to assess changes in zooplankton community composition over time. A non-metric multidimensional scaling analysis (NMDS) analysis was used to assess the pairwise similarities in monthly (June, July, August) zooplankton community composition for each lake [34]. Monthly averages of dominant zooplankton taxa were analyzed using a ranked similarity matrix based on Bray-Curtis similarity measures. Average monthly atmospheric temperatures from the UAF SNAP model were used to add environmental vectors to ordination plots to determine whether changes in zooplankton community composition were associated with warming climate conditions.

### Juvenile sockeye salmon and resident fish

Juvenile sockeye salmon and their freshwater competitors ('resident fish') were sampled annually between 10 August – 10 September via townet in Chignik Lake and Black Lake from 1961−2023. Nets were towed between two boats at the lake surface at a constant speed of approximately 3 km h$^{-1}$. Sampling was conducted at night to reduce net avoidance and capture fish during vertical diel migration [29]. Tow durations varied from 5–10 minutes. Since 1992, tow duration has been standardized to 10 min. In Chignik Lake, a net with a 2 x 2 m opening was used. Declining lake volume made areas of Black Lake inaccessible to gear in 2003 [32]. Prior to 2003, a net with a 1.8 x 1.8 m opening was used, while a net with 1.2 x 1.2 m opening has been used since. Five sites were sampled in each lake, although the outlet site of Black Lake was abandoned in 2003 due to dense macrophyte growth. Since 2003, an additional site in the deeper portion of Black Lake was added to annual surveys.

Prior to 2005, fish were preserved in 10% formalin or 50–70% ethanol and measured at least 24 h after capture. Since 2005, fish were euthanized in a buffered MS-222 solution, held on ice, and measured within 24 h. Length measurements of preserved samples were not corrected following recommendations of Shields and Carlson [35]. If catches were large, a subsample containing at least 100 sockeye salmon was retained for measurement and the sample fraction recorded. Fork length measurements, to the nearest millimeter, were taken from up to 50 individuals of each species from each tow. If a sample contained more than 50 individuals of a species, remaining individuals were enumerated. Juvenile sockeye salmon lengths were standardized to 1 September assuming a growth rate of 0.3 mm per day [36]. For years where multiple nights of towing were conducted, all data were included for analysis. Catch per unit effort (CPUE) for each tow was calculated for juvenile sockeye salmon and three dominant species of resident planktivorous fishes (pond smelt, three-spine stickleback, nine-spine stickleback) as number of fish per m$^2$ of net per min. Rare species (comprising <1% of total catch) such as juvenile coho salmon (*Oncorhynchus kisutch*), coastrange sculpin (*Cottus aleuticus*), and dolly varden (*Salvelinus malma*) were removed prior to analysis. Annual mean CPUE was calculated for two groups, juvenile sockeye salmon and combined resident fish species. These data were converted to log space to normalize residuals. Anomaly plots were used to highlight temporal trends in CPUE for juvenile sockeye salmon and resident fish in both lakes.

Mean annual lengths were calculated for juvenile sockeye salmon retained during townet sampling. In Black Lake, fish larger than 100 mm were removed from the sample as these were rare and assumed to be smolts from earlier spawning events. These fish represent occasional 2 + parr which did not emigrate downstream after their first summer of growth, and comprised less than 0.01% of our sample. In Chignik Lake, multiple stocks and age classes of juvenile sockeye salmon are present. These include two age classes from the late run stock, and one age class from the early run stock which migrate from Black Lake into Chignik Lake during mid-summer [37]. Without available genetic stock identification data, we were unable to definitively assign individuals to known stocks. Visual inspection of annual length distributions revealed two

modes at approximately 50 mm and 70 mm in most years. A hierarchical mixture model was unsuccessful at accurately identifying multiple distributions in years where distributions overlapped or where there was insufficient data. Based upon the known life histories, the smaller mode was assumed to represent Chignik Lake 0+ individuals, while the larger was comprised of both Chignik Lake 1+ and Black Lake 0+ emigrant sockeye salmon [38]. We separated Chignik Lake samples accordingly, calculating annual means for individuals between 45–59 mm and for 60–90 mm. While we cannot assess stock-specific growth rates from this aggregation of the data, we can assess growth performance of the stock aggregates during their freshwater residency. We applied a generalized additive model (GAM) to each sample group to characterize trends in juvenile sockeye salmon length between 1960–2024. For each sample group, we modeled standard length as a smooth function of year. Models were fit using the mgcv package in R [39].

## Results

### Trends in lake temperature

Mean monthly summer water temperatures have remained relatively stable in Black (Fig 2a) and Chignik (Fig 2b) lakes from the 1920s to 2023. Black Lake is slightly warmer (mean temp °C = 11.3, 13.0, 13.0 for June, July and August) than Chignik Lake (mean temp °C = 8.2, 10.7, 12.1). In all years, the average monthly lake temperature in Black Lake and Chignik Lake never exceeded 17.0 °C. The three warmest years on record were 1967, 2019, abd 2002 in Black Lake and 2020, 2016, and 2005 in Chignik Lake. Visual trends suggest mean monthly temperatures have increased minimally in both lakes since 1990, yet these trends were not statistically significant in either lake (all $p > 0.20$; S1 Table). During the freshwater rearing years of the 2018 stock collapse (2013–2016) mean water temperatures were marginally warmer than average in both lakes (mean anomalies were 1.0°C in Black Lake and 1.3°C in Chignik Lake), though were within the range of variation observed earlier in the time series.

Maximum observed monthly water temperatures have increased significantly in Black Lake since 1990 in June, July, and August, with the strongest warming in July (slope = 0.22 °C yr$^{-1}$, $R^2$ = 0.56, $p < 0.001$; Fig 3a; S2 Table). Since 2005, 10 years have experienced temperatures above 18°C. In 2013, temperatures exceeded 18°C in all three summer months. Chignik Lake maximum observed temperatures have increased significantly in June and July, although substantially

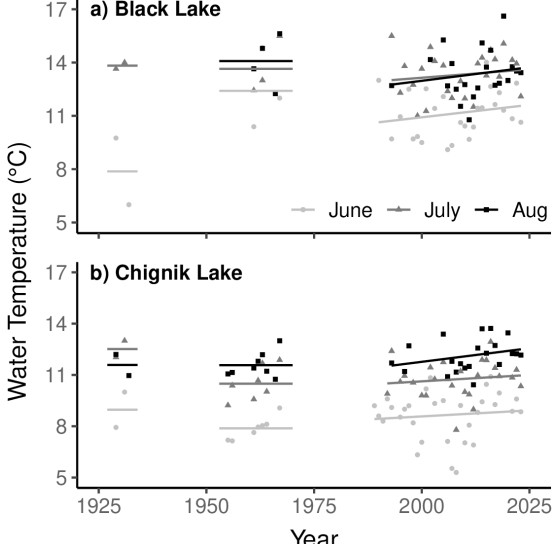

**Fig 2. Mean monthly water temperatures during the summer in Black Lake (a) and Chignik Lake (b) from 1929-2024.**

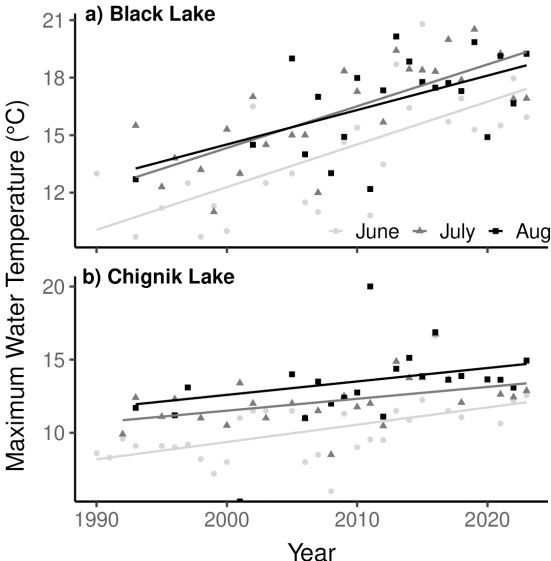

**Fig 3. Maximum observed water temperatures during three summer months in Black Lake (a) and Chignik Lake (b) from 1990-2024.**

less rapidly than Black Lake (S2 Table). The August trend was not significant (Fig 3b). Temperatures above 18°C were observed only in August 2011.

## Zooplankton community composition, abundance, and climate covariates

The zooplankton community of Black Lake (Fig 4a) is numerically dominated by *Bosmina*, with cyclopoid copepods being the second most abundant taxa. Calanoid copepods comprise a small proportion of catches, and *Daphnia* are virtually absent. Total zooplankton abundances showed a slight increasing trend from 2006–2015, with an anomalous increase during 2022 (Fig 5a). In Chignik Lake, cyclopoid copepods have historically dominated the zooplankton community (Fig 4b). Since the early 1990s, *Daphnia*, *Bosmina*, and calanoid copepod contributions have all increased. Between 2013–2020, increasing *Daphnia* abundances contributed to high zooplankton densities in Chignik Lake (Fig 5b) and their proportional contributions to the community were exceptionally high.

NMDS analysis accurately represented the zooplankton community composition in both Black and Chignik lakes during each month of the summer growing season (Table 1) (stress val. < 0.1). In Chignik Lake, *Daphnia* relative abundance was positively correlated with warmer temperatures throughout the summer growing season in all three months considered. In contrast, cyclopoid copepod abundance was correlated with cooler temperatures. In Black Lake, atmospheric temperatures were significantly correlated with the zooplankton community composition during July, with *Bosmina* dominance associated with warmer conditions and cyclopoid copepod with cooler conditions (Fig 6).

Since the freshwater rearing years of the stock collapse (2014–2023), zooplankton abundances in both lakes were on average 230 and 56 individuals/m² above average in Black and Chignik Lakes, respectively. In Chignik Lake *Daphnia* contributions and abundances were notably high between 2016–2020.

## Fish community composition

Juvenile sockeye salmon (42% annual mean composition, 1960–2013) and threespine stickleback (28% annual mean composition, 1960–2013) have historically dominated the fish community of Black Lake (Fig 7a). Between 1992–2024,

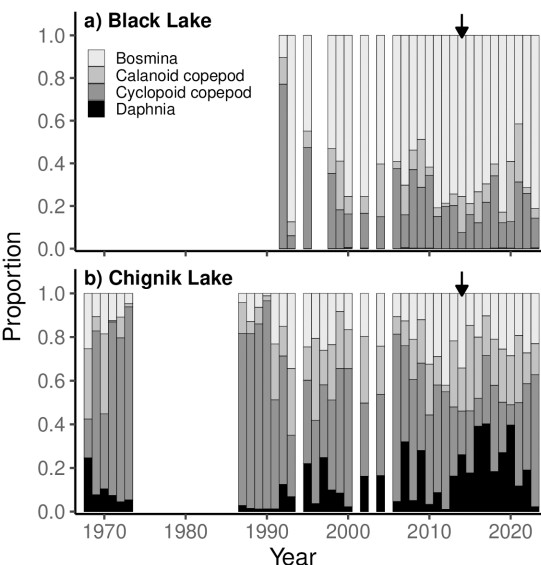

**Fig 4. Annual zooplankton community composition in Black and Chignik lakes.** Black arrows show 1st rearing year of fish that returned as adults during the 2018 sockeye salmon stock collapse.

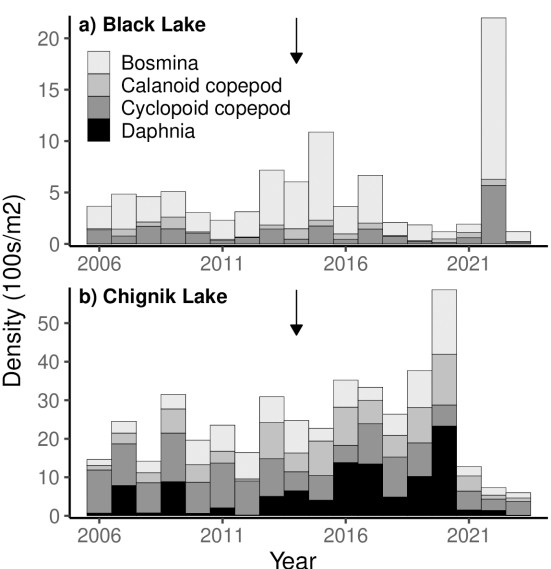

**Fig 5. Annual mean zooplankton densities in Black and Chignik lakes.** Black arrows show 1st rearing year for fish that returned as adults during the 2018 sockeye salmon stock collapse.

pond smelt and ninespine stickleback contributions increased, while threespine stickleback and sockeye salmon contributions were more variable. A notable decrease in sockeye salmon dominance occurred between 2013–2020, during the period of time when the 2018 and subsequent adult returns would have been rearing in the lake. In Chignik Lake, juvenile sockeye salmon historically dominated the resident fish community (78% annual mean composition, 1960–2013) (Fig 7b). Concurrent with the observations in Black Lake, sockeye salmon contributions were notably low between

**Table 1. Comparative summary of zooplankton community NMDS statistics.**

| Lake | Month | NMDS Stress value | *Atmospheric temperature $R^2$ | *Atmospheric temperature P – value |
|------|-------|-------------------|--------------------------------|-------------------------------------|
| Chignik Lake | June | 0.0340 | 0.385 | 0.001** |
| | July | 0.0964 | 0.268 | 0.016** |
| | August | 0.0995 | 0.444 | 0.001** |
| Black Lake | June | 0.0255 | 0.230 | 0.055 |
| | July | 0.0237 | 0.299 | 0.042** |
| | August | 0.0261 | 0.121 | 0.379 |

NMDS, Non-Metric Dimensional Scaling analysis

* Environmental vector

** statistically significant (p < 0.05)

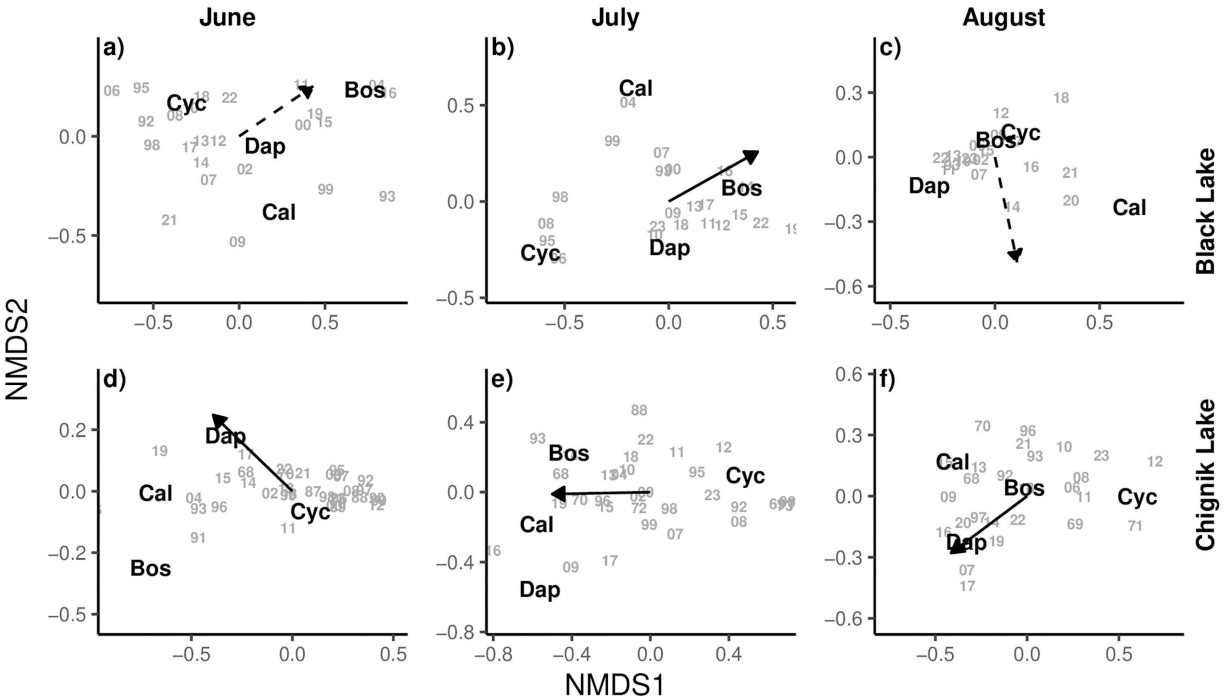

**Fig 6. Non-metric Multidimensional Scaling (NMDS) plot of dominant zooplankton taxa in Black Lake (a, b, c) and Chignik Lake (a, b, c) by month.** Vectors represent an association between atmospheric temperatures and community composition. Stress values for all models were <0.1. Solid vectors show months when temperature had a significant effect (p < 0.05) on species composition. Dashed vectors show months when temperature had no significant effect (p > 0.05) on species composition. Zooplankton taxa are abbreviated as *Daphnia* (Daph), *Bosmina* (Bos), calanoid copepod (Cal), and cyclopoid copepod (Cyl).

2014–2021. Threespine sticklebacks remained the dominant resident species observed in Chignik Lake across the time series.

CPUE anomalies depict an inverse relationship between resident species and juvenile sockeye salmon abundances. Sockeye salmon catches declined sharply between 2017–2021 and 2014–2020 in Black and Chignik Lakes respectively, while resident fish catches increased concurrently (Fig 8).

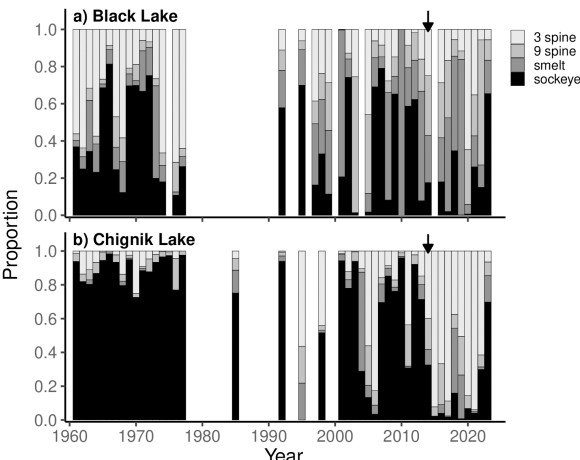

**Fig 7. Annual community composition of dominant planktivorous fish species.** (threespine stickleback, ninespine stickleback, pond smelt, sockeye salmon) in Black (a) and Chignik (b) Lakes. Black arrows show 2014, 1st rearing year for fish that returned as adults during the 2018 collapse.

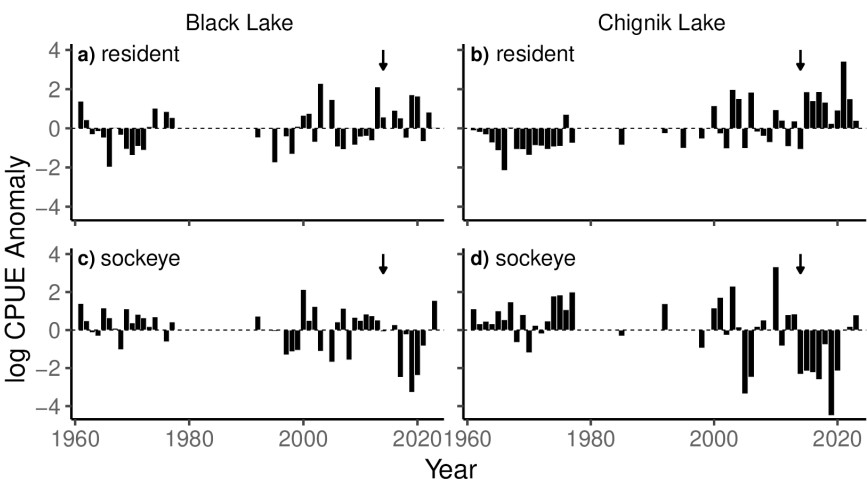

**Fig 8. Log catch per unit effort (CPUE) anomalies for juvenile sockeye salmon and resident fish in Black Lake (a, c) and Chignik Lake (b, d).** Black arrows show 2014, 1st rearing year for fish that returned as adults during the 2018 collapse.

## Juvenile sockeye salmon body size

In Black Lake, mean juvenile sockeye salmon lengths in late August increased between 1978–2020 (Fig 9a). During the freshwater rearing years of the 2018 stock collapse (2014–2016), individuals were on average 9 mm longer than the long-term average. In Chignik Lake, lengths of fish between 45–59 mm have remained relatively consistent, although showed greater interannual variability after 2000 (Fig 9b). Lengths for this group did not appear to decline between 2013–2020. Average lengths of the larger size group (60–90 8kmm) increased between 2000–2021 (Fig 9c). From 2013–2016, individuals were on average 1.7 mm shorter for the small group and 1.6 mm longer in the large group.

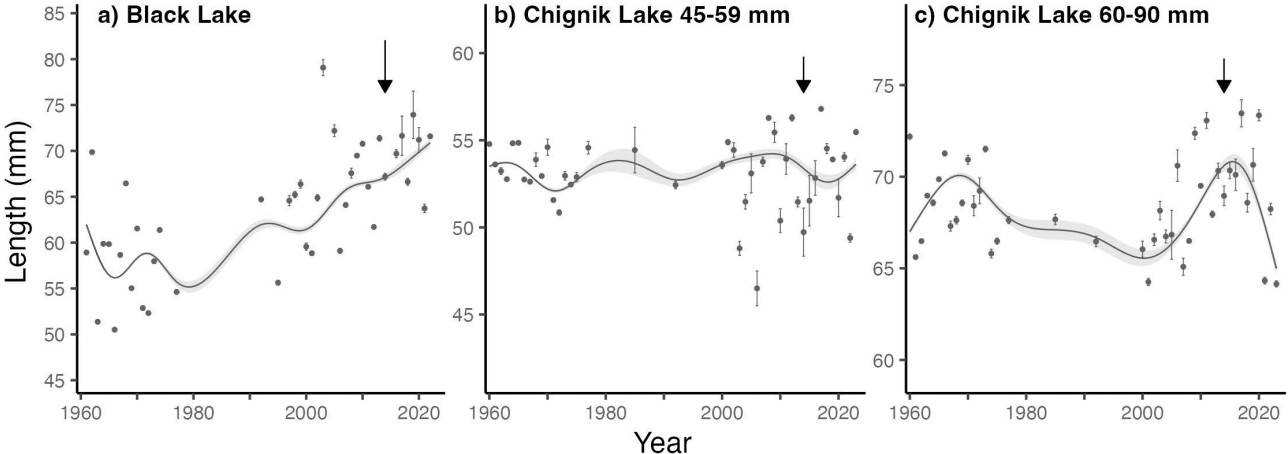

**Fig 9. Juvenile sockeye lengths from townet sampling for Black Lake (a), Chignik Lake 45–60 mm (b), and Chignik Lake 60–90 mm (c).** GAM fit to mean annual lengths, with shaded 95% CI's. Black arrows show 2014, 1st rearing year for fish that returned as adults during the 2018 collapse.

## Discussion

We provide a case study of the long-term changes in the limnological conditions and ecological responses in two lakes that serve as the spawning and nursery habitats for sockeye salmon which have supported valuable commercial and subsistence fisheries for over a century.

We observed high variability in the habitat conditions of both lakes over the last ~60 years, yet did not find evidence for a decline in the overall rearing capacity of the watershed. Considering thermal conditions and prey resources remain conducive for freshwater sockeye salmon growth, it is unlikely that poor habitat conditions for juvenile rearing drove the observed decline in juvenile sockeye salmon abundance observed in 2013 and 2014. The recruit/spawner ratio shows a strong decline in sockeye recruitment beginning in 2013. This suggests either failed adult spawning or reduced egg maturation during the 2013–2016 brood years resulted in a period of low sockeye abundances in freshwater rearing habitat. Collectively, our observations of low sockeye recruitment and freshwater abundance correlate temporally with adults returning during the 2018 collapse. This supports an alternate, yet untested, hypothesis that low early life survival contributed to the collapse observed in 2018 and beyond.

### Water temperature

We identified only subtle trends of increasing water temperatures in both Black and Chignik Lakes since 1995. While changing field methodology precluded us from making a quantitative assessment of long-term changes in thermal conditions in Black and Chignik lakes, the data we have compiled show no large magnitude changes in monthly average water temperatures from the 1920s to the present. It is well documented that climate warming is occurring most rapidly at high latitudes, with similar lakes showing trends of earlier ice-out dates, lengthening growing seasons and increasing summer water temperatures [36,40]. In the Chignik watershed however, consistently strong winds drive mixing of the water column and prevent thermal stratification, thereby buffering against dramatic lake temperature variability driven by trends in atmospheric warming [41]. However, we did find a significant increase in maximum observed summer temperatures in the shallow Black Lake. Although convective cooling buffers average temperatures, even short periods of low wind can produce rapid warming in Black Lake due to its shallow depth and turbid water efficiently absorbing solar radiation.

The distinct geomorphological characteristics of both lakes control local thermal dynamics, resulting in Black Lake remaining consistently warmer than Chignik Lake throughout the summer growing season. Additionally, the shallow depth of Black Lake results in greater temperature variability than Chignik Lake. Chignik Lake has experienced the highest temperature increases during August, likely enhanced by occasional stratification events during the late summer. Stratification of the water column in Chignik Lake remains a rare occurrence, requiring sustained periods of high solar radiation and low wind. As atmospheric temperatures continue to increase, stratification may become more frequent [42]. If the physical structure of Chignik Lake were to stabilize for a significant period of the summer growing season, it is possible new zooplankton communities would emerge producing bottom up behavioral and growth changes for juvenile sockeye salmon [40,43].

As ectotherms, juvenile sockeye salmon bioenergetics are directly controlled by thermal conditions [44,45]. The stock specific juvenile life history strategies exhibited by sockeye salmon in the Chignik watershed reflect the unique thermal regimes of the two lakes. Black Lake juveniles, rearing in a relatively warmer habitat, typically require only one year of freshwater growth. Conversely, Chignik Lake juveniles, rearing in a cooler habitat, typically exhibit two years of freshwater growth. Juvenile sockeye salmon have been shown to experience optimal growth in water temperatures of approximately 15°C, while actively avoiding temperatures >18°C [46]. Our results suggest both lakes remain, on average, near or below the thermal optima for juvenile sockeye salmon growth. In Black Lake, occasional spikes in temperatures exceed 18°C (average n = 6.7 days/year with a maximum temperature > 18°C, 1995–2023), producing temporarily stressful thermal conditions for juvenile sockeye salmon. Typically occurring in the late summer, these events have been hypothesized drivers behind a trend of earlier downstream emigration of poor body condition Black Lake individuals [26,38]. Chignik Lake rarely exceeds 18°C in the epilimnion, and hypolimnetic thermal refugia used by vertically migrating juvenile sockeye salmon persists throughout even the warmest periods [47]. While our data show average temperatures in the Chignik Lakes watershed are slowly rising, the climate and geomorphologic characteristics of the watershed buffer against dramatic changes in water temperatures, and the thermal regimes of both lakes remain productive for juvenile sockeye salmon growth.

During the freshwater rearing years of individuals returning during the 2018 stock collapse, average water temperatures were only slightly warmer in both lakes. In Black Lake however, maximum observed water temperatures exceeded the thermal tolerance for juvenile sockeye salmon at least once in all rearing years of the collapse. While it is likely these episodic stressful thermal conditions contribute to downstream emigration choices of juvenile sockeye salmon in Black Lake, it is unlikely they had a substantial influence on juvenile mortality during the rearing years of the stock collapse because downstream Chignik Lake remains a hospitable and productive rearing habitat prior to their migration to the ocean [16,38].

## Zooplankton community

Our analysis of long-term changes in the zooplankton community found an increase in *Bosmina* dominance in Black Lake beginning in the 1990s, and increasing *Daphnia* contributions in Chignik Lake since 1992 with notably high contributions between 2013–2023 when the fish that returned in 2018 were rearing in the lakes. Additionally, we identified temperature as a control on zooplankton community structure, particularly in Chignik Lake where *Daphnia* dominance was significantly correlated with warmer air temperatures. The four dominant taxa of crustacean zooplankton in the Chignik watershed provide integral ecosystem services as consumers of phytoplankton and as prey resources for planktivorous fish communities. With distinct life histories, zooplankton community structure fluctuates throughout the growing season in response to both bottom-up environmental effects and top-down predator controls. Our results provide evidence of both processes occurring within the watershed.

In Black Lake, *Bosmina* dominance emerged following the decline in lake volume of the 1970s. We found *Daphnia* to be rare within Black Lake, comprising < 0.077% of catches. As large bodied zooplankters, *Daphnia* are a highly preferred

prey resource for planktivores, particularly by juvenile sockeye salmon [48]. With high planktivorous fish abundances in Black Lake and no deep-water predation refuge, size selective predation likely exerts strong top-down controls, limiting *Daphnia* populations [49,50]. Diet studies in Black Lake have shown juvenile sockeye salmon are able to achieve rapid growth without access to *Daphnia* by exploiting abundant invertebrate prey resources, primarily chironomid larvae [29].

With a deeper water column, cooler temperatures, and lower densities of planktivorous fish, the zooplankton community of Chignik Lake is not only more diverse, but also more dynamic than Black Lake. Since 1992, *Daphnia* composition has tended to increase. Employing a NMDS analysis, we were able to test for correlation between climate variables and zooplankton community composition. We observed a significant correlation between atmospheric temperatures and zooplankton community composition across the growing season within Chignik Lake. Notably, years with warmer temperatures were associated with greater *Daphnia* contributions. As *Daphnia* have a competitive and reproductive advantage at higher temperatures, this result was expected and supports the hypothesis that climate variables exert control on zooplankton communities through bottom-up processes [40,51]. Additionally, cyclopoid copepods, which experience peak abundances in early spring, were more dominant during cooler years.

The distinct increase in *Daphnia* contributions between 2013–2023 may also suggest top-down predation controls on the zooplankton community within Chignik Lake. Beginning in 2013, low sockeye salmon catch rates correlate temporally with the increased *Daphnia* contributions we observed. Released from high levels of sockeye salmon predation, *Daphnia* were likely able to exploit their competitive advantages resulting in increased dominance in the zooplankton community. This response suggests that juvenile sockeye abundances may have been reduced for these year classes, earlier in their life cycles; for example, either from failed adult spawning or from poor egg-to-fry survival rates.

## Planktivorous fish community

In the planktivorous fish communities of both Black and Chignik Lakes, we observed a dramatic decline in juvenile sockeye salmon dominance since 2013 and 2014, respectively. Additionally, our results suggest an inverse relationship of sockeye salmon and resident species abundances throughout the watershed.

Previous studies have shown sockeye salmon and resident fish populations can effectively coexist within Black Lake due to high prey densities and discrete habitat preferences [52]. Black Lake sockeye rely heavily on invertebrate prey, allowing gape limited stickleback to exploit abundant zooplankton populations without significant interspecific competitive pressures [29]. While sockeye salmon have historically dominated the fish community of Black Lake, Westley et al. (2008) documented an increase in resident fish abundances following the geomorphological shifts of the 1970s. They concluded lower lake levels had improved climate mediated stickleback recruitment, increasing competitive pressures on small sockeye unable to target invertebrate prey. Facing intra- and interspecific competitive pressures, and increased thermal stress, poor condition individuals were observed to migrate downstream earlier in the season seeking habitat refugia [26]. The declines in sockeye dominance and catch rates we observed suggest this process has continued and intensified. While the thermal conditions and prey resources of Black Lake enable rapid growth of fit individuals, it is likely poor condition sockeye continue to emigrate downstream earlier in the season [38]. This helps explain the high resident fish abundances and low juvenile sockeye salmon catch rates of the last decade.

With cooler water temperatures, lower productivity, and minimal littoral spawning habitat, resident fish species are at a competitive disadvantage to sockeye salmon in Chignik Lake. Historically, juvenile sockeye salmon have successfully exploited these favorable habitat conditions and strongly dominated the fish community. As downstream emigration of Black Lake sockeye salmon intensifies, we would expect to see a corresponding increase in juvenile sockeye salmon densities in Chignik Lake. Our results contradict this hypothesis, showing juvenile sockeye salmon dominance and catch rates fell sharply between 2014–2021. While climate mediated recruitment is likely increasing resident fish abundances in Chignik Lake, there is limited littoral rearing habitat compared to Black Lake. Additionally, warming temperatures and higher *Daphnia* abundance during this period suggest the pelagic habitat quality of Chignik Lake has improved for juvenile sockeye salmon since 2013. These observations support the hypothesis of high early life stage mortality, observed in the

 

poor recruitment of 2012–2016 sockeye brood years. With low sockeye salmon abundance, resident fish were relieved from historically high competitive pressures, enabling rapid population growth.

### Juvenile sockeye salmon growth

Juvenile salmon body length reflects the habitat mediated growth performance of individuals during freshwater residence [53]. Our analysis of juvenile sockeye salmon lengths showed increasing growth trends in Black Lake, and stable growth trends in multiple Chignik Lake size classes. These observations refute the hypothesis of significant declines in habitat quality across the watershed, yet alone do not directly reflect lake specific rearing capacity. While habitat conditions remain conducive for juvenile sockeye salmon growth, changing emigration patterns resulting in increased population mixing complicate our interpretation of lake specific rearing capacity [38]. Further, stock-specific spawning success would also obscure clear patterns in stock-specific growth patterns in Chignik Lake as density-dependent growth would be responsive to both intra-stock competition and inter-stock competition within the watershed.

For mobile species in interconnected ecosystems, source-sink dynamics emerge as individuals experience stressors in source habitats [37]. The ability of individuals to migrate in search of habitat refugia can buffer total species abundances from localized habitat change, leading to decreased abundances in source habitats and novel population interactions in sink habitats [16]. As we have described, Black Lake is shown to act as a source habitat for juvenile sockeye within the Chignik watershed [37]. A trend of poor condition individuals emigrating downstream earlier in the summer has likely decreased interspecific competitive pressures on sockeye salmon remaining in Black Lake for the entire growing season. With access to high quality habitat and less competition, our results of increasing length trends suggest individuals remaining in Black Lake have experienced improved rearing conditions since the 1980s. Notably, this trend continued through the collapse rearing years of 2014–2016, refuting the hypothesis that a decline in Black Lake habitat quality contributed to the 2018 stock collapse.

Behaving as a potential sink habitat, Chignik Lake supports both natal and emigrant sockeye populations. Juvenile sockeye salmon exhibit strong density dependence, suggesting population mixing likely drives asynchronous growth responses for individual age classes and stocks [36,54]. With multiple mixed stocks and age classes, a cumulative length analysis of all Chignik Lake catches does not specifically reflect population specific growth performance. Previous research has successfully solved this issue using genetic stock identification techniques to assign individuals to either Black or Chignik Lake populations [38,55]. Without available genetics data for our entire time series, our analysis was limited to describing growth performance of two stock aggregates, both of which showed stable growth trends. These results suggest that any changes in the habitat quality of Chignik Lake have not had a negative effect on juvenile sockeye salmon growth. Additionally, juvenile sockeye salmon growth during the brood years of the 2018 collapse remained consistent with growth during historic periods of high adult returns.

Relying on the coarse stock aggregation methodology we employed has several drawbacks. Crucially, it does not account for stock specific responses to evolving density dependent pressures, driven by the climate mediated downstream migration of Black Lake individuals [26]. However, Griffiths et al. (2013) observed that Black Lake emigrants did not exhibit significantly different body condition to natal Chignik Lake juveniles [38]. This suggests that while there may be more Black Lake origin individuals rearing in Chignik Lake, the emigrants are not at a significant competitive advantage over their Chignik Lake counterparts. In the decade since these studies were conducted, it remains unknown if increased Black Lake emigration has influenced natal Chignik Lake juvenile sockeye salmon growth performance.

### Conclusion

Variation in habitat conditions within and among watersheds supports reliable and sustainable salmon fisheries in Alaska where ecosystems remain largely unperturbed from development [10,28]. Long term changes in geomorphology and climate drive shifts in habitat structure, organism interactions, and population performance [32]. Here we have shown that multiple components of the Chignik lakes meta-ecosystem are highly variable, reflecting the dynamic nature of

interconnected heterogeneous watersheds. Black and Chignik lakes exhibit divergent habitat characteristics, with asynchronous responses by juvenile sockeye salmon stocks to habitat change. Despite high variability, our assessment of rearing conditions between 2013–2016 found no evidence that freshwater habitat was less profitable for juvenile sockeye salmon during the rearing years that led to the 2018 stock collapse. Notably we observed abundant zooplankton resources throughout the watershed, and an increase in more profitable prey species in Chignik Lake. Juvenile sockeye salmon growth performance was above average, despite increasing resident fish populations. It is likely lower sockeye salmon abundances have decreased density-dependent pressures, enabling individuals to exploit improved prey resources and achieve faster growth. Additionally, the low juvenile sockeye salmon abundances we observed between 2013–2014 suggest a population bottleneck during the early life stages that likely contributed to the disastrous returns of 2018. Although our analysis was conducted with comprehensive long term datasets, investigating numerous proxies for freshwater rearing capacity, it is possible that habitat features which support juvenile rearing underwent changes we were unable to measure with available data.

While our hydrologic data are incomplete, several winter floods that occurred in 2013 and onwards may have resulted in poor egg survival in the major spawning tributary of Black Lake due to gravel scour. Such poor recruitment of sockeye salmon fry into Black and Chignik lakes is apparent both in our estimates of abundance, in indirect indicators of their predation pressure (i.e., the zooplankton community composition), and may have propagated to produce failed year classes that led to the fishery collapses in 2018 and beyond.

Our analysis focused exclusively on the freshwater habitat conditions of juvenile sockeye salmon, yet changes in estuarine and marine habitat may have also contributed to the 2018 stock collapse. From 2013–2017, anomalously high sea surface temperatures in the North Pacific resulted in dramatic changes to marine food webs [56,57]. Increasing marine temperatures resulted in a higher diversity of lower quality zooplankton prey species available to adult salmon [58]. Cary et al. (2021) observed a decline in sockeye salmon spawning success correlated with the marine heatwave, suggesting poor marine prey resources reduced the number of viable eggs deposited [59]. Last, declining body sizes of sockeye salmon as has been documented in other parts of Alaska [60,61] may also be contributing to reduced 'spawner quality' which may need to be incorporated into revising escapement goals to improve sockeye salmon yield from the Chignik watershed [62]. These other hypotheses warrant investigation and should be considered as potential alternative explanations for the recent collapse of Chignik sockeye salmon fisheries. Further research on stock specific growth performance, juvenile sockeye life history strategies, and winter flood events in particular would aid in contextualizing our findings by potentially identifying mechanisms behind the 2018 collapse. While our results do not support adjusting current management strategies, testing these alternative hypotheses would benefit managers seeking to develop strategies to avoid future stock collapses. For fisheries managers tasked with maintaining population diversity and productivity of a mixed stock fishery, this study highlights the stability of freshwater rearing capacity of the Chignik Lakes watershed, and the importance of interconnected heterogeneous habitat networks in buffering long term populations.

## Supporting information

**S1 Table. Linear regression statistics for mean monthly surface water temperatures.**
(DOCX)

**S2 Table. Linear regression statistics for maximum monthly surface water temperature.**
(DOCX)

## Acknowledgments

Jackie Carter processed zooplankton samples and provided invaluable data management support. Eli Fournier, Jonathan Singleton, Nicholas Chambers, Ben Makhlouf, and Emma Christman were among the many UW employees and volunteers who assisted with sample collection as part of our long-term project in the Chignik watershed. Curry

Cunningham and Mark Scheuerell provided advice on statistical approaches and comments on the manuscript. This is a contribution of the University of Washington Alaska Salmon Program.

## Author contributions

**Conceptualization:** Cirque Gammelin, Daniel E Schindler.

**Data curation:** Cirque Gammelin.

**Formal analysis:** Cirque Gammelin.

**Funding acquisition:** Daniel E Schindler.

**Investigation:** Cirque Gammelin.

**Methodology:** Cirque Gammelin.

**Project administration:** Daniel E Schindler.

**Resources:** Daniel E Schindler.

**Supervision:** Daniel E Schindler.

**Visualization:** Cirque Gammelin.

**Writing – original draft:** Cirque Gammelin.

**Writing – review & editing:** Cirque Gammelin, Daniel E Schindler.

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
