## [Decision Letter · Decision Letter 0]

24 Nov 2025

PONE-D-25-52212Long-term Changes in the Juvenile Sockeye Salmon Rearing Capacity of the Chignik Lakes WatershedPLOS ONE

Dear Dr. Gammelin,

Thank you for submitting your manuscript to PLOS ONE. After careful consideration, we feel that it has merit but does not fully meet PLOS ONE’s publication criteria as it currently stands. Therefore, we invite you to submit a revised version of the manuscript that addresses the points raised during the review process. Specifically, the reviewers have provided generally positive comments on the manuscript; however, some criticisms have been made on some minor points which have to be thoroughly addressed in a revised version of the ms before it can be further considered for publication in PLOS ONE. Please submit your revised manuscript by Jan 08 2026 11:59PM. If you will need more time than this to complete your revisions, please reply to this message or contact the journal office at plosone@plos.org. Please include the following items when submitting your revised manuscript:

We look forward to receiving your revised manuscript.

Kind regards,

Giorgio Mancinelli, Ph.D.

Academic Editor

PLOS ONE

Journal Requirements:

“Funding was provided by the Gordon and Betty Moore Foundation (CG,DS), the Chignik Regional Aquaculture Association (DS), the Fishery Disaster Relief Program of the Pacific States Marine Fisheries Commission (DS), and the School of Aquatic and Fishery Sciences at the University of Washington (CG, DS).”

“Funding was provided by the Gordon and Betty Moore Foundation, the Chignik Regional Aquaculture Association, the Fishery Disaster Relief Program of the Pacific States Marine Fisheries Commission, and the School of Aquatic and Fishery Sciences at the University of Washington.”

“Funding was provided by the Gordon and Betty Moore Foundation (CG,DS), the Chignik Regional Aquaculture Association (DS), the Fishery Disaster Relief Program of the Pacific States Marine Fisheries Commission (DS), and the School of Aquatic and Fishery Sciences at the University of Washington (CG, DS).”

Additional Editor Comments (if provided):

The reviewers have provided on the manuscript generally positive comments, even though criticisms have been made on some minor points which have to be thoroughly addressed before the ms can be further considered for publication in PLOS ONE

Reviewers' comments:

Reviewer's Responses to Questions

**Comments to the Author**

1. Is the manuscript technically sound, and do the data support the conclusions?

Reviewer #1: Yes

Reviewer #2: Partly

2. Has the statistical analysis been performed appropriately and rigorously? 

Reviewer #1: Yes

Reviewer #2: Yes

3. Have the authors made all data underlying the findings in their manuscript fully available?

Reviewer #1: Yes

Reviewer #2: No

4. Is the manuscript presented in an intelligible fashion and written in standard English?

Reviewer #1: Yes

Reviewer #2: Yes

5. Review Comments to the Author

Reviewer #1: This is a well-written manuscript that uses several multi-decal datasets to determine if Sockeye salmon rearing capacity in the Chignik Lakes watershed has declined. The authors have done an excellent job at utilizing existing long-term data and assessing whether the physical and biological conditions in Black Lake and Chignik lakes have changed to indicate a loss in rearing capacity preceding the 2018–2021 stock collapse. I appreciate the authors well-organized and crafted paragraphs as well as the numerous figures to display the results. The manuscript is suitable for publication in PLOS ONE and I strongly recommend acceptance for publication.

Please see my minor comments, suggestions, and questions attached to the PDF. Also, figure 3 appears to be missing from my PDF. Could this be a technical issue caused by the manuscript submission page? Is there a figure or upload limit?

Reviewer #2: Thank you for the opportunity to review this manuscript. I thoroughly enjoyed reading it and believe that with revision it will make a valuable contribution to understanding factors contributing to Pacific salmon declines.

My overarching comments relate mainly to how the research is presented to ensure the data support the conclusions presented (publication criteria #4) and clear presentation of ideas and the broader context of the work (publication criteria #5).

Overall, the manuscript is well-written in an intelligible fashion. However, as someone with a general knowledge of Pacific salmon fisheries but not specific knowledge of the Chignik region, I found the work could be better placed in the broader context of sockeye salmon fishery declines in the region, including both freshwater and marine factors that could contribute to declines. For example, the potential role of harvest and interactions with species that prey on sockeye salmon eggs/juveniles are not currently discussed. The manuscript appears to make a case for the hypothesis that warmer temperatures in Black Lake led individuals in poor condition to emigrate downstream early to Chignik Lake, where littoral rearing habitat is limited, leading to high early life stage mortality. However, assessing emigration/origins of individuals is not possible. Other hypothesis such as failed spawning, low egg maturation and survival are mentioned (mainly later in the conclusion) and the data presented don’t support the hypothesis that declines in rearing habitat quality primarily caused the 2018 collapse. I recommend the authors revise to provide a broader background on the Chignik sockeye fishery collapse and potential contributing factors before narrowing in on the impacts of climate on rearing habitat. I further recommend aligning how findings are presented in the manuscript text with how they are presented in the abstract, which does so more objectively. The discussion would benefit from leading with main, overarching findings and not following the same structure as the results to convey relationships between indicators more clearly and concisely.

Unfortunately, they dryad link provided did not work and therefore I was unable to confirm whether all data underlying findings are fully available. I was curious what spawner and recruit data were used to estimate recruits/spawner and how these data were collected. The figures appear out of order/mislabeled and I believe that Fig. 3 might be missing? Lastly, I recommend a thorough read to screen for typos (e.g., lines 318, 351, 575, 594).

Attached are specific comments in a pdf. I hope the authors find these comments helpful for improving their manuscript.

Best,

Jenilee Gobin

6. PLOS authors have the option to publish the peer review history of their article (what does this mean?). If published, this will include your full peer review and any attached files.

Reviewer #1: **Yes:** Jason C. Leppi

Reviewer #2: **Yes:** Jenilee Gobin

You may also use PLOS’s free figure tool, NAAS, to help you prepare publication quality figures: https://journals.plos.org/plosone/s/figures#loc-tools-for-figure-preparation

---

## [Author Response · Author response to Decision Letter 1]

26 Jan 2026

Dear PLOS One Editorial Board,

We greatly appreciate the feedback provided by reviewers and have made the following revisions to our manuscript. Below we describe our responses to the comments provided by the reviewers and changes made to the manuscript.

In response to comments on our funding statement, we have removed funders from the acknowledgement section of the manuscript. A new funding statement can be found in the updated cover letter.

In response to the comment concerning the map in Figure 8 (now Figure 1), we have replaced the original figure with a newly generated map created entirely in R using publicly available spatial data. The shaded relief was independently generated in R from the USGS 3DEP DEM and therefore constitutes a new derived work. Hydrography data were obtained from the USGS National Hydrography Dataset, coastline data were obtained from NOAA, and the Alaska inset map was created using Natural Earth data.

Revisions made to reviewer comments by line number:

[47] Moved sentence to beginning of paragraph to begin Introduction with climate change > lake ecosystems > population responses

[63] Added citation (Quinn, 2018)

[96] Added sentence in figure 1 caption indicating 2018 stock collapse year and escapement goals.

[97] Added escapement goals to figure 1.

[101] Added “Comparing trends in rearing capacity between interconnected shallow (Black lake) and deep (Chignik lake) nursery lakes, we investigated asynchronous changes to habitat and population dynamics in a heterogeneous watershed.”

[104] Added corresponding to the typical 1-2 years of freshwater residency and 2-3 years of marine residency of sockeye salmon.” To clarify life cycle timing, and also provided relevant citations (Narvar 1966, Dhalberg 1979).

[249] Removed duplicate description of juvenile sockeye salmon and resident fish spp. CPUE calculation.

[288] Simplified results statement by removing mean summer temp results for Chignik lake. The increases were small, and complicated interpretation of key result that evidence for rapid temperature increases is lacking. Replaced with ; “Visual trends suggest mean monthly temperatures have increased minimally in both lakes since 1995, yet these trends were not statistically significant in Black (p > 0.05) or Chignik (p > 0.2) lake.”

[308] Figure 3 has now been uploaded.

[318] Moved figure reference to the end of the sentence and corrected capitalization.

[328] We believe Figure 5 should remain in the manuscript body as it highlights the increasing abundance (not just proportion) of Daphnia in Chignik lake, thereby providing additional evidence of improved growth opportunity for juvenile salmon in the ecosystem.

[340] Added ‘Non-metric Multidimensional scaling (NMDS)’

[341] Replaced ‘influence’ with ‘an association between atmospheric temperatures and community composition.’

[340] Added ‘Dashed vectors show months when temperature had no significant effect (p > 0.05) on species composition.’

[351] Added ‘,’ before ‘respectively’

[398] See line 509.

[419] Added; “These short term increases in water temperature can drive juvenile sockeye salmon to seek thermal refugia downstream in Chignik Lake (26).”

[431] This is supported by Carter & Schindler 2012 and Berger et al. 2010. I replaced ‘likely’ with ‘possible’.

[443] Clarified sentence.

[450] Edited paragraph for clarity

[502] We highlight the evidence that there was less predation on the zooplankton community in this lake, possibly indicating reduced recruitment of juvenile sockeye salmon into the lake. However, to quantify the relative effects of changing predation and climate variables on zooplankton community composition would require data that we do not have in our work. Thus, we have retained this statement as a likely explanation for the dynamics we observed in the zooplankton community but we are not able to further explore it in this paper.

[509] Moved paragraph proposing early life stage mortality event to discussion introduction.

[528] Added citation

[530] Added citation. Previous studies (Westley et al. 2008, Griffiths et al. 2013) in the watershed have identified earlier downstream emigration during warmer years.

[555] We agree that this statement is speculative and have reworded to clarify that changes in emigration patterns will obscure changes in stock-specific spawning success in each of the nursery lakes. Thus, we added the sentence. ‘Further, stock-specific spawning success would also obscure clear patterns in stock-specific growth patterns in Chignik Lake as density-dependent growth would be responsive to both intra-stock competition and inter-stock competition within the watershed.’

[563] Added citation

[611] We believe this discussion belong late in the Discussion because it is a hypothesis that is generated from the analyses presented above.

[629] Added “Our results highlight the importance of interconnected heterogeneous habitat networks in buffering long term populations. Further research on stock specific performance and juvenile life history strategies of Chignik sockeye salmon would help inform management strategies aimed at maintaining population diversity long term productivity of a mixed stock fishery.”

We thank the reviewers for their thoughtful and constructive feedback, which has improved the clarity of the manuscript. We hope that our revisions satisfactorily address all comments, and we welcome any further suggestions.

Sincerely,

Cirque Gammelin

---

## [Decision Letter · Decision Letter 1]

10 Mar 2026

PONE-D-25-52212R1Long-term Changes in the Juvenile Sockeye Salmon Rearing Capacity of the Chignik Lakes WatershedPLOS One

Dear Dr. Gammelin,

Thank you for submitting your manuscript to PLOS ONE. After careful consideration, we feel that it has merit but does not fully meet PLOS ONE’s publication criteria as it currently stands. Therefore, we invite you to submit a revised version of the manuscript that addresses the points raised during the review process.

We look forward to receiving your revised manuscript.

Kind regards,

Giorgio Mancinelli, Ph.D.

Academic Editor

PLOS One

Journal Requirements:

Additional Editor Comments:

The authors made a successful attempt to incorporate the comments of the reviewers in the revised version of the manuscript, to the point that one of the reviewer consider it acceptable for publication. However, the second reviewer highlighted some additonal minor points to be addressed in an additional revision round before the ms can be further considered for publication in PLOS One

Reviewers' comments:

Reviewer's Responses to Questions

**Comments to the Author**

1. If the authors have adequately addressed your comments raised in a previous round of review and you feel that this manuscript is now acceptable for publication, you may indicate that here to bypass the “Comments to the Author” section, enter your conflict of interest statement in the “Confidential to Editor” section, and submit your "Accept" recommendation.

Reviewer #1: All comments have been addressed

Reviewer #2: (No Response)

2. Is the manuscript technically sound, and do the data support the conclusions?

Reviewer #1: Yes

Reviewer #2: Partly

3. Has the statistical analysis been performed appropriately and rigorously? 

Reviewer #1: Yes

Reviewer #2: Yes

4. Have the authors made all data underlying the findings in their manuscript fully available?

Reviewer #1: Yes

Reviewer #2: Yes

5. Is the manuscript presented in an intelligible fashion and written in standard English?

Reviewer #1: Yes

Reviewer #2: Yes

6. Review Comments to the Author

Reviewer #1: (No Response)

Reviewer #2: Thank you for the opportunity to review this revised manuscript. The authors have incorporated line by line feedback well. However, I suspect they might have missed a comment related to providing readers with a broader background on potential factors contributing to the Chignick sockeye fishery in the introduction before focusing on the potential role of changes in rearing habitat. Just 1-2 lines could be added in the paragraph starting on line 82 to address this, where the potential role of floods could also be mentioned briefly. Additionally, some details (e.g., reporting statistics) and typos/grammatical errors appear to have been overlooked. The implications of the study’s findings for management and future research needed for management summarized in the conclusion could also be more specific and clearly articulated.

I hope the authors find these additional comments helpful to continue improving their manuscript.

Best,

JL

Line by line comments:

- some typos/grammatical errors remain in the manuscript (e.g., line 71 missing comma between ‘degraded’ and ‘juvenile’, line 86 missing the word ‘of’ between ‘years’ and ‘painful’)

- year 2020 on x-axis cut off in Fig 1c

- year intervals in lines 186-187 don’t align with Fig 3

- linear regression analyses should report additional statistics beyond p values (e.g., test statistics, correlation coefficient)

- line 352 (previously 340) referring to defining abbreviations in caption was in reference to the abbreviated prey species depicted in the figure (i.e., listing prey species in the caption is more explicit and helps the figure ‘standalone’)

- paragraph starting line 461 (previously 433) – lack of clarity stemmed from paragraph structure. The message the reader is intended to take away is not immediately clear. What is the main point this paragraph aims to convey?

- lines 521-530 (previously 493-502) – suggest ‘couching’ text in this paragraph as was done in the response provided; present as a possible indication of top down effects as the authors clearly stated in their response that the data needed to determine mechanisms driving trends in zooplankton communities are not available. I don’t necessarily disagree with your claim that changes in zooplankton could reflect a release from predation; I’m simply suggesting you present it as a possibility rather than suggesting the evidence presented is sufficiently strong to infer mechanistic/causal relationships.

- line 585 (previously 555-557) – do not see additional sentenced referenced in the author’s response in the revised text

Conclusion:

- The current study does not suggest a decline in rearing habitat based on patterns in juvenile growth, temperature, zooplankton, and fish communities. This study assessed proxies for habitat but was unable to quantify rearing habitat directly – it remains possible that habitat features that support spawning and juveniles underwent changes that could not be measured in this study that contributed to failed spawning/low juvenile survival.

- The hypothesis that floods in 2013 could have contributed to the 2018 collapse does not emerge from the current study. I understand the authors focused on the question of potential declines in rearing habitat as a potential factor contributing to declines. They could nonetheless introduce this alternate hypothesis earlier. The alternative hypotheses presented in the conclusion should be moved into the introduction so that the conclusion can highlight the specific study implications and the questions it raises. It would be more useful to readers (and managers specifically) to present the broader context at the outset and situate the question of the role of rearing habitat within this.

- The introduction frames the study in the specific context of providing a comprehensive analysis to support effective management thus warranting more specific conclusions/recommendations for management in its conclusion. In light of your findings, is the proposed/current management strategy supported or not? It would help to inform the readers of what management actions are currently being taken and identify specific research is needs for adaptive management based on the current findings.

7. PLOS authors have the option to publish the peer review history of their article (what does this mean?). If published, this will include your full peer review and any attached files.

Reviewer #1: **Yes:** Jason C. Leppi

Reviewer #2: **Yes:** Jenilee Gobin

You may also use PLOS’s free figure tool, NAAS, to help you prepare publication quality figures: https://journals.plos.org/plosone/s/figures#loc-tools-for-figure-preparation

---

## [Author Response · Author response to Decision Letter 2]

25 Apr 2026

Dear PLOS One Editorial Board,

We greatly appreciate the feedback provided by reviewers and have made the following revisions to our manuscript. Below we describe our responses to the comments provided by the reviewers and changes made to the manuscript.

Revisions made to reviewer comments, by line number:

[68] & [83] fixed grammatical errors throughout

[Fig 1c] Fixed year 2020 on x-axis

[95] added paragraph to introduction addressing 2nd conclusion comment. I intend this paragraph broaden background on other factors which may contribute to a stock collapse, and introduce alternate hypotheses earlier in the text.

‘With diverse and complex life histories, it can be challenging to identify and quantify the mechanisms behind sockeye salmon stock dynamics. Factors such as freshwater habitat quality, oceanic conditions, recruitment success, and harvest all contribute to the temporal stability of individual stocks [27,28]. While this study focuses on the hypothesized declines in freshwater rearing capacity in the Chignik watershed, it is possible multiple mechanisms contributed to the 2018 stock collapse. For example, an emerging hypothesis asks if extreme early winter flood events may have scoured sockeye salmon redds, resulting in poor egg and fry survival in the years preceding the collapse.’

[183] (previously 186) corrected temperature sampling year groups.

[301-325] – reported additional test statistics and edited results for clarity. Added S1 and S2 Tables [733]. Added sentence in methods [197],

[350] (previously 325) added “Species abbreviations for dominant zooplankton taxa are; Daphnia (Daph), calanoid copepod (Cal), cyclopoid copepod (Cyc), and Bosmina (Bos)” to Figure 6 caption.

[449] (previously 461) I can see what you mean. This paragraph intends to describe the relationship between temperature and sockeye bioenergetics in the watershed, how the life histories of both stocks reflect the thermal regimes of the two lakes, and to state the consistent and productive thermal conditions in both lakes.

I edited the entire ‘water temperature’ section of the discussion for clarity, removing duplicate statements and hopefully improved overall clarity. Notable changes below, see markup copy for all changes.

[435] removed sentence repeated in line 457

[448] added sentences describing the connection between freshwater life history strategies and lake specific thermal regimes.

[458] removed ‘Not only are smaller sockeye salmon at a competitive foraging disadvantage, they may also be less resilient to stressful thermal regimes.’

[467] added “While our data show average temperatures in the Chignik Lakes watershed are slowly rising, local climate and geomorphologic characteristics likely buffer against dramatic changes in water temperatures, and the thermal regimes of both lakes remain productive for juvenile sockeye salmon growth.”

[534] Rephrased paragraph to present observations as a possible explanation rather than evidence.

[565] (previously 585-557) added missing sentence

‘Further, stock-specific spawning success would also obscure clear patterns in stock-specific growth patterns in Chignik Lake as density-dependent growth would be responsive to both intra-stock competition and inter-stock competition within the watershed’

[659] added sentence addressing 1st conclusion comment.

‘Although our analysis was conducted with comprehensive long term datasets, investigating numerous proxies for freshwater rearing capacity, it is possible that habitat features which support juvenile rearing underwent changes we were unable to measure with available data.’

[689] Response to 3rd conclusion comment on management context

- The proposed management changes (described in the introduction) were revoked before this study concluded (line 89), and there have been no management changes since. The point of introducing the proposed management changes was to introduce the hypothesis of the study.

- I agree that it is necesary to acknowledge the importance of our results for informing future management, but only in the context of providing evidence of a resilient habitat network capable of supporting multiple productive sockeye salmon stocks and suggesting further research.

- See changes made in last paragraph of conclusion.

We thank the reviewers for their thoughtful and constructive feedback, which has improved the clarity of the manuscript. We hope that our revisions satisfactorily address all comments, and we welcome any further suggestions.

Sincerely,

Cirque Gammelin

---

## [Editor Report · Decision Letter 2]

27 Apr 2026

Long-term Changes in the Juvenile Sockeye Salmon Rearing Capacity of the Chignik Lakes Watershed

PONE-D-25-52212R2

Dear Dr. Gammelin,

We’re pleased to inform you that your manuscript has been judged scientifically suitable for publication and will be formally accepted for publication once it meets all outstanding technical requirements.

Kind regards,

Giorgio Mancinelli, Ph.D.

Academic Editor

PLOS One

Additional Editor Comments (optional):

In this second revision of the manuscript, all the minor points raised by the reviewers have been successfully addressed, making the ms acceptable for publication in PLOS One
---

## [Editor Report · Acceptance letter]

PONE-D-25-52212R2

PLOS One

Dear Dr. Gammelin,

I'm pleased to inform you that your manuscript has been deemed suitable for publication in PLOS One. Congratulations! Your manuscript is now being handed over to our production team.

Kind regards,

on behalf of

Dr. Giorgio Mancinelli

Academic Editor

PLOS One